# Meaning in Life, Social Axioms, and Emotional Outcomes during the First Outbreak of COVID-19 in Hong Kong

**DOI:** 10.3390/ijerph20136224

**Published:** 2023-06-25

**Authors:** Rong-Wei Sun, Esther Yuet Ying Lau, Sing-Hang Cheung, Chi-Keung Chan

**Affiliations:** 1School of Arts and Humanities, Tung Wah College, Hong Kong; irissun@twc.edu.hk; 2Department of Psychology, The Education University of Hong Kong, Hong Kong; 3Department of Psychology, The University of Hong Kong, Hong Kong; singhang@hku.hk

**Keywords:** meaning in life, social axioms, negative emotions

## Abstract

Social unrest, coupled with the outbreak of COVID-19, was a double-hit for Hong Kong in early 2020. Those stressful societal situations not only trigger negative emotions, such as anxiety and/or depression, but also consolidate a person’s belief towards oneself (i.e., meaning in life) and society (i.e., social axioms). The study included 2031 participants from the Formation and Transformation of Beliefs in Chinese (FTBC) project dataset. The data were collected in Hong Kong from February 2020 to March 2020 (double-hit). Path analysis and multiple regression were used to examine the mediating and moderating effects of the presence subscale (P) of the Meaning in Life Questionnaire (MLQ) on the relations between social axioms and negative emotions. Results showed that low MLQ-P mediated the associations between cynicism and negative emotions and between low religiosity and negative emotions and moderated the relation between social cynicism and emotional outcomes. Exploratory analyses showed that MLQ-Search (S) mediated the relations between reward for application and negative emotions, between social complexity and negative emotions, and between fate control and negative emotions, and moderated the relation between religiosity and stress. As far as we know, this study reported the first evidence of the role of meaning in life in explaining and modifying the associations between social axioms and mood states. The presence of and search for meaning in life seem to work differently with respect to the relations between social axioms and negative emotions, with important implications for understanding the dynamics of social and personal beliefs in affecting mental health in times of large-scale public crisis.

## 1. Introduction

Following the first confirmed case on 23 January 2020, Hong Kong experienced a severe impact from the first and second waves of COVID-19. The pandemic, as well as the multiple anti-pandemic strategies including intense surveillance for infection, mandatory quarantine for inbound travelers, social distancing, school closures, etc., led to significant disruptions to residents’ daily lives [1]. Meanwhile, the COVID-19 outbreak arrived with the ending of the social protest in Hong Kong [2]. Conceivably, such a double-hit of COVID-19 and social unrest could induce tremendous psychological impact. A study reported prevalences of 19% and 14% for moderate–severe depression and anxiety symptoms, respectively [3]. A few studies have examined the roles of different variables in modulating the impacts of COVID-19 and/or social unrest, such as demographic risk factors (e.g., low socioeconomic status) and social variables (e.g., social support) [2]. To shed light on contributors to community mental health during crises, it is essential to further investigate the individual differences that underlie psychological outcomes under this double-hit. 

The personal mechanism this on which study focuses is meaning in life, which refers to how one makes sense of, and attributes significance to, one’s existence and being [4]. Steger et al. [4] used factor analysis to identify two factors in meaning in life: presence of meaning in life (MLQ-P) and search for meaning in life (MLQ-S). The former measures the subjective experience of meaning in life, whereas the latter measures the process of finding and looking for meaning in life. Of note, studies have found positive associations between psychological well-being and meaning in life when it was measured as one holistic construct [5,6]; however, more divergent effects have been reported, with positive associations with the presence of meaning in life and a negative correlation with its search [7]. Steger and his colleagues proposed that the negative correlation might be explained by uncertainties towards one’s life during the search for meaning in life [7]. 

Many studies reported a mediating effect of meaning in life on the relations between adverse events and psychological outcomes. For instance, Arslan et al. [8] found that psychological maltreatment predicted lowered psychological well-being through reducing meaning in life (indirect effect: 0.08, *se* = 0.03, *bootCI* = [0.02, 0.15]). Some studies showed that only the presence mediated the relationship between negative events and psychological outcomes. For example, Kelly et al. [9] showed that the low presence (indirect effect: 0.08, *CI* = [0.03, 0.14]), but not the lack of search (indirect effect: −0.002, *CI* = [−0.02, 0.01]), explained the association between moral injury and suicide ideation for former military members. Meanwhile, some studies showed that both the presence of and the search for meaning in life mediated the relationship between negative events and psychological outcomes. Lew et al. [10] showed that both presence and search mediated the relation between negative focus and suicidal behavior (presence indirect effect: 0.050, *CI* = [0.009, 0.067]; search indirect effect: 0.014, *CI* = [0.006, 0.022]); both mediated the relationship between suicide orientation and suicidal behaviors (presence indirect effect: 0.048, *CI* = [0.007, 0.061]; search indirect effect: 0.016, *CI* = [0.005, 0.025]) and between suicide orientation and suicidal behaviors (presence indirect effect: 0.048, *CI* = [0.007, 0.061]; search indirect effect: 0.016, *CI* = [0.005, 0.025]); however, the link between hopelessness and suicidal behaviors was only significantly mediated by the presence of meaning (indirect effect: 0.061, *CI* = [0.028, 0.077]), but not by the search for meaning (the path from search to suicidal behaviors was insignificant, β = −0.038, *p* > 0.05). One recent study in Hong Kong showed that both presence and search mediated the relation between future orientation and prosocial tendency (indirect effect: β = 0.141, *p* < 0.001), and the paths from presence to prosocial tendency and search to prosocial tendency were significantly positive (presence β = 0.227, *p* < 0.001 and search β = 0.220, *p* < 0.001) [11]. 

On the other hand, meaning in life, and mostly its presence, was found to moderate the relation between adverse events and psychological outcomes. For example, one study by Appel et al. [12] showed that the presence attenuated the relations between goal violation and post-traumatic stress (PTS) symptoms (*b* = −0.17, *p* < 0.001), and between religious/spirituality struggles and PTS symptoms (*b* = 0.18, *p* = 0.015). Similarly, other studies showed the mitigating effects of meaning in life on the links between hopelessness and suicide risk, between eating attitude and suicide ideation, and between eating attitude and hopelessness [13,14], although null moderating effects have also been reported [15]. 

The social mechanism we tested in this study was individuals’ generalized beliefs towards the social world, referred to as the social axioms [16]. There are five dimensions of social axioms: social cynicism; social complexity; reward for application; spirituality; and fate control [17]. First, social cynicism is defined as a generalized mistrust towards humans and society. People with high scores in social cynicism tend to believe that other people are self-centered, manipulative, and only want to take advantage of others. Earlier research suggests that social cynicism is linked to negative outcomes such as lower life satisfaction [18], increased internet gaming addiction [19], and psychological distress and emotional rumination [20]. Second, religiosity refers to a generalized belief in the positive effects of religious practices and in the existence of a supreme being. Studies showed that religiosity positively correlated with life satisfaction [21] and psychological well-being [22]. Interestingly, meaning in life seemed to buffer the negative effects of low spiritual wellness on depression [23]. Another study provided evidence from a serial mediational model from spiritual experiences to hope, then to meaning in life, and lastly to life satisfaction [24]. (The two studies [23,24] imply that the relations between religiosity and negative emotions could either be moderated or mediated by meaning in life). 

To date, relatively few studies have been conducted on the relations between the other three social axioms (social complexity, reward for application, and fate control) and meaning in life due to the small effect size and conflicting results varying from one paper to another. Social complexity—defined as a generalized view on the variability of individual behaviors across situations and the multiplicity of outcomes and their pathways [18]—has been shown to be associated with emotional instability [20] and life dissatisfaction [25]. Reward for application, referred to as a generalized belief on hard work and effort, was found to be positively associated with subjective well-being [21]. People with high fate control believe in predetermined destiny over one’s own control of his/her life. Studies showed small to negligible effect sizes between fate control and life satisfaction [21,25]. It is important to note that while the existing literature provides some insights into these three social axioms, there still remain many unanswered questions that warrant further research. 

Given the possible associations among social axioms, meaning in life, and negative emotion outcomes, this study aimed to investigate the mediating or moderating effects of meaning in life on the associations between social axioms and negative mood states. Considering the previous literature on the mediating effect of the presence of meaning in life [8,10,11], this study proposed the following hypotheses (as shown in Figure 1a):

**Hypotheses** **1** **(H1).**
*Low presence would mediate the relation between social cynicism and negative emotional outcomes.*


**Hypotheses** **2** **(H2).**
*Low presence would mediate the relation between religiosity and negative emotional outcomes.*


Based on previous studies that showed the moderating effect of presence of meaning in life [12,13,14], this study also proposed the following hypotheses (as shown in Figure 1b):

**Hypotheses** **3** **(H3).**
*Presence would attenuate the relation between social cynicism and negative emotional outcomes.*


**Hypotheses** **4** **(H4).**
*Presence would buffer the relation between low religiosity and negative emotional outcomes.*


Moreover, to gain a more comprehensive understanding of the connections among the other three social axioms (i.e., reward for application, social complexity, and fate control) and the search for meaning in life with negative mood states, we included these variables in both the mediation and moderation models. In these models, the social axioms were the exogenous variables in path models or predictors in regression, the presence of and the search for meaning in life were the mediators or moderators, and the negative emotions were the endogenous variables in path models or outcomes in regression. 

**Figure 1 ijerph-20-06224-f001:**
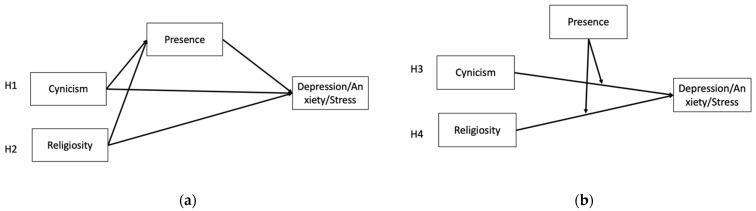
Hypothesized models: mediated (**a**) and moderated (**b**) effect of the presence of meaning in life.

## 2. Methods

### 2.1. Participants

The data were collected during the first outbreak of COVID-19 in Hong Kong from February to March 2020. Participants were recruited through a longitudinal study database, the Formation and Transformation of Beliefs in Chinese (FTBC), as well as through social media advertisements and mass emails in local universities. The Human Research Ethics Committee of the co-corresponding author’s institution has approved the data collection procedures. A total of 2031 adults who lived in Hong Kong completed this survey. 

### 2.2. Measures

The Meaning in Life Questionnaire (MLQ), developed by Steger et al., was used to measure the meaning in life [4]. Respondents rated 10 items on a 7-point Likert scale, ranged from 1 “Absolutely untrue” to 7 “Absolutely true”. It contains two subscales: presence of meaning in life (MLQ-P) and search for meaning in life (MLQ-S). The Chinese version of MLQ showed good reliabilities (Cronbach’s α = 0.85 for both subscales) and good model fit indices with a bifactor model [10]. In this study, this questionnaire also showed good reliability (presence: α = 0.90 and search α = 0.89). 

Social Axioms (SAS-II) were measured by 25 items, adapted from Leung et al.’s original 44-item scale [16]. SAS-II’s adopted 5-point Likert scale ranged from 1 = “Strongly disagree” to 5 = “Strongly agree”. There are five subscales with five items for each subscale: social cynicism; reward for application; social complexity; fate control; and religiosity. The Cronbach’s α values for the five subscales in this study were as follows: social cynicism (0.62); reward for application (0.72); social complexity (0.59); fate control (0.67); and religiosity (0.81). Previous studies have shown that this questionnaire has acceptable reliabilities ranging from 0.64 to 0.79, which are quite close to this study [16]. This questionnaire was adopted mainly because its items were derived from Hong Kong participants and media sources [16]. 

The study used the 21-item Depression, Anxiety, Stress Scale (DASS) [26] to measure the negative emotion outcomes. This scale is composed of three subscales: depression; anxiety; and stress. Each subscale consists of 7 items, rated on a 4-Likert scale (0 = “did not apply” to 3 = “applied most of the time”). A higher score for each subscale represents severe symptoms. The Cronbach’s alpha in this study for depression, anxiety, and stress were 0.87, 0.86, and 0.79, respectively.

Demographic characteristics include gender, age, educational attainment, religion, and monthly family income. 

### 2.3. Statistical Analysis

All statistical analyses were implemented using R script (version 4.2.2) [27]. Descriptive analyses were performed for all study variables, including mean, standard deviation, range, skewness, and kurtosis. We also reported Cronbach’s alpha and Omega coefficients. A Cronbach’s alpha of 0.70 and a MacDonal’s Omega of 0.50 would be regarded as acceptable [28]. 

Path analyses were used to examine if the presence of and search for meaning in life can explain the link between social axioms and negative emotions. The three path models were tested with the R package, lavaan [29]. The missing data were handled with full information maximum likelihood (FIML). The bootstrapped confidence intervals were computed by the R package, semhelpinghands [30]. The 95% bootstrapping confidence intervals for the standardized path coefficients were obtained with 5000 bootstraps resamples by the delta method using the variance–covariance matrix of bootstrap estimates. 

The moderation effects of presence of and search for meaning In life were examined by the three multiple regressions using the R package, stdmod [31]. For ease of interpretation, all the variables were standardized before forming the product terms, as recommended by Cheung et al. [31]. We also used the non-parametric bootstrapping suggested by Cheung et al. [31], with 5000 resamples to compute the confidence interval of the regression coefficients for the moderations. 

## 3. Results

### 3.1. Descriptive Characteristics

Table 1 summarized the demographic characteristics of the respondents. Overall, the mean age of the participants was 22.91 (*SD* = 3.33, ranging from 18 to 35 years old; *n* = 1398, as only 1398 out of 2031 reported their age). Most participants were women (74.24%, *n* = 1507), 59.77% were college students (*n* = 1214), the rest of the participants (*n* = 816)—around 87.99% (*n* = 718 out of 816)—had a full-time job, and around 95.22% (*n* = 777 out of 816) had obtained a college degree or above. Most participants identified as Atheist (70.01%). Nearly half of them had a monthly family income higher than 30,000 HKD (55.59%, *n* = 1129). 

The descriptive statistics, Cronbach’s alpha, and McDonald’s omega for study variables are shown in Table 2. The correlations among the study variables are shown in Table 3.

### 3.2. Path Models: Mediating Effects of Meaning in Life on Negative Emotions (Hypotheses 1 and 2)

The first model (presented in Figure 2a and Table 4) showed social cynicism positively related to depression (β = 0.187, *p* < 0.001, 95% *bootCI* [0.147, 0.226]). The indirect path of social cynicism to depression through presence of meaning in life was significant (β = 0.073, *p* < 0.001, 95% *bootCI* [0.053, 0.094]), but the indirect path through search for meaning in life was non-significant (β = −0.004, *p* > 0.05, 95% *bootCI* [−0.010, 0.001]). The direct path from religiosity to depression was non-significant (β = −0.023, *p* > 0.05, 95% *bootCI* [−0.062, 0.016]). The indirect path of religiosity to depression through presence of meaning in life was significant (β = −0.054, *p* < 0.001, 95% *bootCI* [−0.075, −0.034]), but not through search for meaning (β = 0.004, *p* > 0.05, 95% *bootCI* [−0.001, 0.009]). 

The second model (presented in Figure 2b and Table 4) showed social cynicism positively related to anxiety (β = 0.214, *p* < 0.001, 95% *bootCI* [0.169, 0.258]). The indirect path of social cynicism to anxiety through presence of meaning in life was significant (β = 0.035, *p* < 0.001, 95% *bootCI* [0.024, 0.049]), but the indirect path through search for meaning in life was non-significant (β = −0.005, *p* > 0.05, 95% *bootCI* [−0.012, 0.001]). The direct path from religiosity to anxiety was non-significant (β = −0.002, *p* > 0.05, 95% *bootCI* [−0.044, 0.042]). The indirect path of religiosity to anxiety through presence of meaning in life was significant (β = −0.025, *p* < 0.001, 95% *bootCI* [−0.038, −0.016]), but not through search for meaning (β = 0.004, *p* > 0.05, 95% *bootCI* [−0.001, 0.011]). 

The third model (presented in Figure 2c and Table 4) showed social cynicism positively related to stress (β = 0.149, *p* < 0.001, 95% *bootCI* [0.145, 0.234]). The indirect path of social cynicism ro stress through presence of meaning in life was significant (β = 0.028, *p* < 0.001, 95% *bootCI* [0.024, 0.049]), but the indirect path through search for meaning in life was non-significant (β = −0.004, *p* > 0.05, 95% *bootCI* [−0.011, 0.000]). The direct path from religiosity to stress was non-significant (β = 0.001, *p* > 0.05, 95% *bootCI* [−0.043, 0.043]). The indirect path of religiosity to stress through presence of meaning in life was significant (β = −0.020, *p* < 0.001, 95% *bootCI* [−0.039, −0.016]), but not through search for meaning (β = 0.003, *p* > 0.05, 95% *bootCI* [−0.039, 0.048]). 

To sum up, our results supported H1 and H2, i.e., that the presence of meaning in life partially mediated the relations between social cynicism and emotional outcomes. It also fully mediated the relations between religiosity and negative emotional outcomes. All the estimates of the path coefficients (including unstandardized path coefficients, standard error of unstandardized path coefficients, standardized path coefficients, and the bootstrapping confidence intervals) are shown in Appendix A, Table A1. 

However, we could not find evidence for the search for meaning in life mediating the associations between cynicism and emotional outcomes, nor those between religiosity and emotional outcomes. 

### 3.3. Multiple Regression Model: Moderating Effects of Meaning in Life on Negative Emotions (Hypotheses 3 and 4)

The presence of and search for meaning in life were treated as multiple moderators on the relationships between social axioms and depression. The moderating effects were examined by multiple regressions, as shown in Table 5. When the outcome was depression, the multiple regression analysis showed that predictors including presence and search for meaning in life, social axioms, and the interaction terms explained 32.40% of the variance in depression (*F*(17, 2010) = 56.54, *p* < 0.001). The interaction term between the presence of meaning and social cynicism was significant (cynicism*presence *b* = −0.071, *se* = 0.019, *p* < 0.001, 95% *bootCI* = [−0.109, −0.031]). The regression coefficient for the search for meaning were non-significant (*b* = −0.009, *p* > 0.05), and no interaction effects were found for search for meaning. To clarify the moderating effect of the presence of meaning in life in the relation between social cynicism and depression, simple slope analyses for high (+1 *SD*), moderate, and low (−1 *SD*) levels of presence of meaning in life were conducted (*b* = 0.124, *p* < 0.001, 95% *bootCI* [0.079, 0.170]), (*b* = 0.194, *p* < 0.001, 95% *bootCI* [0.155, 0.236]), and (*b* = 0.265, *p* < 0.001, 95% *bootCI* [0.201, 0.330]), respectively. For anxiety, there was no interaction effect. All the predictors without interaction terms explained 16.40% of the variance in anxiety (*F*(17, 2009) = 23.19, *p* < 0.001). 

For stress, 15.64% of its variance was explained by all the predictors including interaction terms (*F*(17, 2019) = 21.91, *p* < 0.001). The interaction terms between social cynicism and presence of meaning (*b* = −0.062, *p* < 0.01, 95% *bootCI* = [−0.105, −0.019]), between religiosity and presence of meaning (*b* = −0.044, *p* < 0.05, 95% *bootCI* = [−0.085, −0.002]), and between reward for application and search for meaning (*b* = 0.050, *p* < 0.05, 95% *bootCI* = [−0.003, 0.103]) were significant. The simple slope analyses found that social cynicism positively correlated with stress for both low (−1 *SD*), medium, and high (+1 *SD*) presence of meaning in life (low: *b* = 0.258, *se* = 0.032, *p* < 0.001, 95% *bootCI* = [0.184, 0.331]; medium: *b* = 0.196, *se* = 0.022, *p* < 0.001, 95% *bootCI* = [0.149, 0.241]; and high: *b* = 0.134, *se* = 0.030, *p* < 0.001, 95% *bootCI* = [0.082, 0.183]). Simple slope analyses showed that in high (+1 *SD*), medium, and low (−1 *SD*) presence of meaning in life, the associations between religiosity and stress were all non-significant (high: *b* = −0.035, *se* = 0.027, *p* = 0.202, 95% *bootCI* = [−0.082, 0.012]; medium: *b* = 0.010, *se* = 0.022, *p* = 0.655, 95% *bootCI* = [−0.035, 0.055]; low: *b* = 0.054, *se* = 0.032, *p* = 0.087, 95% *bootCI* = [−0.020, 0.127]). Simple slope analyses showed that the relation between reward for application and stress was significant for low search for meaning in life (*b* = −0.123, *se* = 0.031, *p* < 0.001, 95% *bootCI* = [−0.201, −0.044]) but not for high search for meaning in life (*b* = −0.023, *se* = 0.030, *p* = 0.444, 95% *bootCI* = [−0.090, 0.049]). 

To sum up, the results partly supported H3, i.e., that the high presence of meaning in life attenuates the association between social cynicism and depression and between social cynicism and stress. In addition, the results partly supported H4, i.e., that the presence of meaning in life buffered the negative association between religiosity and stress. These results seem to imply a cross-over interaction; however, in the simple slope analyses, all regression coefficients were non-significant, probably due to the weak effect of religiosity on stress (*b* = 0.010, *p* > 0.05). 

## 4. Discussion

The current study investigated the mediating and moderating effects of the presence of meaning in life and explored the role of the search for meaning in life in the relations between social axioms and negative emotions. 

### 4.1. The Role of Presence of Meaning in Life

Previous studies only considered the effect of meaning in life on the link between adverse life events (e.g., trauma) and emotional or behavioral outcomes (e.g., anger or suicidal behavior) [8,9,10]. The current study broadens these indirect effects of meaning in life to the associations between social axioms and negative emotions. Our findings supported H1 and H2, i.e., that the low presence of meaning mediated the relations between social cynicism and negative mood states and between low religiosity and negative mood states. These findings further suggest that young adults with a higher sense of the presence of meaning in life are less likely to hold cynical beliefs and less vulnerable to negative emotional symptoms. Partially in line with previous studies showing buffering effects of the presence of meaning in life on negative psychological outcomes [10,11,12], we also found moderating effects of the presence of meaning in life on the relations between social cynicism and depression, between social cynicism and stress, and between religiosity and stress (simple slope analysis showed that with the high, medium, and low levels of presence, the paths from religiosity to stress were all nonsignificant, probably due to the small effect from religiosity to stress (*b* = 0.010, *p* > 0.05)), partly supporting H3 and H4. The multiple buffering effects of the presence of meaning in life suggest its importance in challenging times. The association between higher levels of social cynicism and more depression and stress differed in size depending on the levels of presence of meaning in life. Higher presence buffers such associations between social cynicism and depression (and/or stress). The results imply that the presence of meaning in life is a protective factor and reduces the influence of social cynicism on negative emotions. 

As suggested by Hui and Hui [32], personal belief and social belief could work together in affecting one’s psychological well-being. This study further extends the idea that personal belief could mediate the association between social belief and negative emotions. Specifically, social beliefs, such as cynicism, might gradually shape one’s personal beliefs, such as meaning in life, which, in turn, influence negative emotions. 

### 4.2. The Role of Search for Meaning in Life

Furthermore, similar to previous studies in which the presence of meaning in life was shown to be positively associated with positive outcomes such as well-being [33] and negatively associated with negative outcomes such as suicide ideation [8,9], this study found that the presence of meaning was correlated with less depression, anxiety, and stress symptoms. In addition, consistent with most previous studies, but somewhat different from some studies including a local one [8,9,11,33], the search for meaning in life was related to more depression, anxiety, and stress symptoms in our model. This could be explained by the presence-to-search model proposed by Steger et al. [6], wherein when people have deficits in meaning (low score on the presence of meaning), they might display a high need to search for meaning. The explanation for the conflicting results of the search could be how the participants understand the search for meaning in life. If the search is driven by a lack of presence, it may be related to more mood symptoms. However, if the search is due to a continuous drive of self-improvement or self-actualization, it may be related to fewer mood symptoms. Such different interpretations of the search for meaning in life could be triggered by social or cultural contexts. 

In addition, the relation between reward for application and stress was moderated by the search for meaning in life. At a high level of search for meaning in life (+1 *SD*), the reward for the application did not significantly relate to stress; at a low level of search (−1 *SD*), the reward for application was negatively related to less stress. A believer in reward for efforts might feel less stress, as one expects to progress or even find significance by working hard. However, for those who are still in search of meaning in life, the stress-relieving effects of obtaining rewards from applying oneself may not be adequate, as the ultimate significance of hard work may still be questionable without a solid grasp of one’s own meaning in life. 

### 4.3. Social Axioms and Meaning in Life

Put together, these results suggest that social axioms and meaning in life could have different pathways to different negative emotions. For instance, one mediation effect found in our exploratory analyses in this study was that of meaning in life (both presence of and search for meaning in life) mediating the relations of fate control with negative emotions. Specifically, people high in fate control related to the low presence of meaning and reported more depression, anxiety, and stress symptoms. Meanwhile, people high in fate control related to a high search for meaning and related to fewer anxiety and stress symptoms. Contrary to some previous studies such as Bachem et al. [3], and supported by other studies such as Kuo et al. [34], this study showed that fate control was associated with higher stress. Fate control is considered as a belief in fixed and pre-planned destiny; people with fatalistic views might hold quite stable meaning in life (presence) and maintain that one’s fate is decided, directed at the fulfillment of the ultimate purpose, and thus one cannot change or control the fate. In addition, our study found fate control correlated with more negative emotions (depression, anxiety, and stress). Under some circumstances, higher fate control might imply the helplessness and desperation that one cannot change anything. Considering that the data in this study were collected during the double-hit in Hong Kong, the negative correlation between fate control and presence might be a sign of the conflict between social beliefs (fate control) and existing personal beliefs (presence). 

### 4.4. Limitations and Future Directions

Several limitations of the present study should be noted. First, we used cross-sectional data. Future studies could collect longitudinal data for examining mediation effects. Second, although our study explored the search and presence of meaning in life, future studies could explore different understandings of the search for meaning in life (i.e., the search could be treated as a lack of presence or as a state of exploring for personal growth) involved in the association between social axioms and negative emotional outcomes. Researchers could also consider adopting mixed research design, which incorporates both quantitative and qualitative methods into the research in order to gain a deeper understanding of meaning in life. Third, future analysis could generalize this model to other samples such as the elder population. 

## 5. Conclusions

Building on the theory of the interactions between personal and societal beliefs and well-being outcomes, this study provided evidence of the mediating and moderating roles of the presence of and search for meaning in life in the relationships between social axioms and negative emotions during a challenging time, namely, the double-hit of the COVID-19 pandemic following social unrest. Overall, the persistently beneficial effect of the presence of meaning in life (personal beliefs) partially explains and also modifies the effects of social axioms (social beliefs), particularly social cynicism on negative emotions. Our exploratory findings on the search for meaning in life provide novel evidence of how the search for meaning in life might be triggered by one’s social belief and indicate that it might be one of the mechanisms through which certain social beliefs may induce negative moods in an individual. Based on this evidence, future longitudinal studies may examine the complex causal associations between personal and social variables and their contribution to well-being outcomes in times of crisis. 

## Figures and Tables

**Figure 2 ijerph-20-06224-f002:**
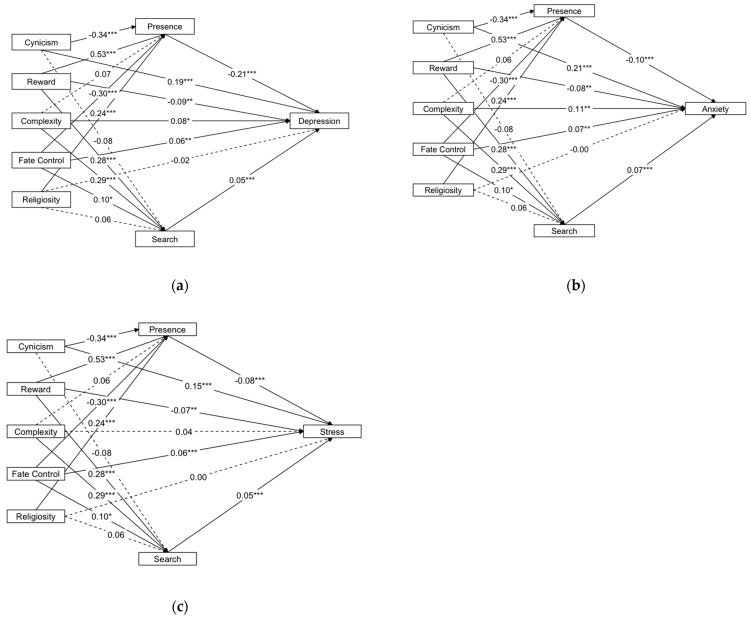
The path model with standardized coefficients. The covariances among social axioms and between presence and search were included but not shown in this figure. The covariance between presence and search was nonsignificant: *b* = 0.056; β = 0.049; *p* = 0.091. The dashed lines represent nonsignificant path coefficients. (**a**) Mediating effects on depression; (**b**) mediating effects on anxiety; (**c**) mediating effects on stress. * *p* < 0.05. ** *p* < 0.01. *** *p* < 0.001.

**Table 1 ijerph-20-06224-t001:** Demographic characteristics of participants.

		*N*	M (SD) or Freq. (%)
Age (year)		1398	22.91 (3.33)
Gender		2030	
	Female		1507 (74.24%)
	Male		523 (25.76%)
Education attainment		2029	
	Secondary		3 (0.37%)
	Upper secondary		1250 (61.61%)
	Diploma/University Degree		499 (24.59%)
	Postgraduate degree		277 (13.65%)
Religion	Atheist	2031	1422 (70.01%)
	Christian		453 (22.30%)
	Catholic		71 (3.50%)
	Buddhist		67 (3.30%)
	Others		18 (0.89%)
Monthly Family income	(HKD)	1926	
	<10,000		102 (5.30%)
	10,000–19,900		295 (15.32%)
	20,000–29,900		400 (20.77%)
	30,000–39,900		371 (19.26%)
	40,000–49,900		229 (11.89%)
	>50,000		529 (27.47%)

*N* = sample size; *M* = mean; *SD* = standard deviation; Freq. = frequency.

**Table 2 ijerph-20-06224-t002:** Descriptive statistics for study variables.

	*k*	*N*	*M*	*SD*	*Range*	*Skew*	*Kurt*	Cronbach’s Alpha	McDonald’s Omega
SAS									
Cynicism	5	2031	2.95	0.62	1–5	−0.03	−0.25	0.62	0.63
Reward	5	2031	3.91	0.51	1.8–5	−0.63	1.32	0.72	0.72
Complexity	5	2031	4.29	0.37	2.4–5	−0.18	0.54	0.59	0.60
Fate Control	5	2030	3.29	0.61	1–5	−0.15	0.03	0.67	0.63
Religiosity	5	2031	3.35	0.64	1–5	−0.24	0.67	0.81	0.69
MLQ									
MLQ-P	5	2030	4.88	1.36	1–7	−0.64	−0.45	0.80	0.80
MLQ-S	5	2031	4.65	1.53	1–7	−0.08	−1.18	0.83	0.83
DASS									
Depression	7	2030	0.71	0.62	0–3	1.21	1.36	0.87	0.87
Anxiety	7	2029	0.88	0.62	0–3	0.75	0.17	0.86	0.85
Stress	7	2029	0.53	0.49	0–3	1.40	2.30	0.79	0.80

*k* = the number of items; *N* = sample size; *M* = mean; *SD* = standard deviation; *Skew* = skewness; *Kurt* = kurtosis.

**Table 3 ijerph-20-06224-t003:** Correlation matrix among the study variables.

	1	2	3	4	5	6	7	8	9	10
1. Cynicism	1									
2. Reward	−0.198 ***	1								
3. Complexity	0.033	0.238 ***	1							
4. Fate Control	0.251 ***	−0.138 ***	0.036	1						
5. Religiosity	−0.174 ***	0.199 ***	0.066 **	−0.078 ***	1					
6. MLQ-P	−0.273 ***	0.301 ***	0.068 **	−0.230 ***	0.212 ***	1				
7. MLQ-S	−0.063 **	0.172 ***	0.137 ***	0.026	0.078 ***	0.100 ***	1			
8. Depression	0.332 ***	−0.222 ***	0.019	0.218 ***	−0.154 ***	−0.505 ***	0.032	1		
9. Anxiety	0.297 ***	−0.148 ***	0.057 *	0.187 ***	−0.088 ***	−0.286 ***	0.081 ***	0.695 ***	1	
10. Stress	0.276 ***	−0.161 ***	0.020	0.188 ***	−0.088 ***	−0.289 ***	0.060 **	0.692 ***	0.746 ***	1

* *p* < 0.05; ** *p* < 0.01; *** *p* < 0.001.

**Table 4 ijerph-20-06224-t004:** Summary of direct and indirect paths in the mediational path models.

	Depression		Anxiety		Stress	
	β	95% *bootCI*	β	95% *bootCI*	β	95% *bootCI*
Direct effect						
Cynicism -> DASS	0.187 ***	[0.147, 0.226]	0.214 ***	[0.169, 0.258]	0.189 ***	[0.145, 0.234]
Reward -> DASS	−0.070 **	[−0.119, −0.021]	−0.068 **	[−0.117, −0.020]	−0.073 **	[−0.126, −0.021]
Complexity -> DASS	0.045 *	[0.004, 0.084]	0.062 **	[0.017, 0.105]	0.027	[−0.020, 0.076]
Fate -> DASS	0.057 **	[0.018, 0.097]	0.071 **	[0.029, 0.115]	0.079 **	[0.033, 0.125]
Religiosity -> DASS	−0.023	[−0.062, 0.016]	−0.002	[−0.044, 0.042]	0.001	[−0.043, 0.043]
Indirect Paths						
Cynicism -> MLQ-P -> DASS	0.073 ***	[0.053, 0.094]	0.035 ***	[0.024, 0.049]	0.036 ***	[0.024, 0.049]
Reward -> MLQ-P -> DASS	−0.093 ***	[−0.117, −0.070]	−0.045 ***	[−0.060, −0.031]	−0.045 ***	[−0.061, −0.032]
Complexity -> MLQ-P -> DASS	−0.008	[−0.026, 0.010]	−0.004	[−0.013, 0.005]	−0.004	[−0.013, 0.005]
Fate -> MLQ-P -> DASS	0.063 ***	[0.043, 0.083]	0.031 ***	[0.020, 0.043]	0.031 ***	[0.020, 0.043]
Religiosity -> MLQ-P -> DASS	−0.054 ***	[−0.074, −0.034]	−0.026 ***	[−0.038, −0.016]	−0.026 ***	[−0.039, −0.016]
Cynicism -> MLQ-S -> DASS	−0.004	[−0.010, 0.001]	−0.005	[−0.012, 0.001]	−0.005	[−0.011, 0.000]
Reward -> MLQ-S -> DASS	0.013 **	[0.006, 0.021]	0.016 ***	[0.008, 0.025]	0.014 **	[0.006, 0.022]
Complexity -> MLQ-S -> DASS	0.009 **	[0.004, 0.017]	0.012 **	[0.005, 0.020]	0.010 **	[0.004, 0.017]
Fate -> MLQ-S -> DASS	0.005	[0.001, 0.011]	0.007 *	[0.001, 0.014]	0.004 *	[−0.001, 0.010]
Religiosity -> MLQ-S -> DASS	0.004	[−0.001, 0.009]	0.005	[−0.001, 0.011]	0.005	[−0.039, 0.048]
Total indirect effect						
Cynicism -> DASS	0.069 ***	[0.048, 0.090]	0.030 ***	[0.017, 0.045]	0.031 ***	[0.019, 0.046]
Reward -> DASS	−0.080 ***	[−0.104, −0.057]	−0.029 **	[−0.046, −0.013]	−0.032 ***	[−0.049, −0.016]
Complexity -> DASS	0.001	[−0.018, 0.020]	0.008	[−0.003, 0.019]	0.006	[−0.004, 0.017]
Fate -> DASS	0.068 ***	[0.048, 0.089]	0.037 ***	[0.024, 0.051]	0.036 ***	[0.024, 0.050]
Religiosity -> DASS	−0.050 ***	[−0.072, −0.029]	−0.021 **	[−0.034, −0.009]	−0.022 ***	[−0.035, −0.011]
Total effects						
Cynicism -> DASS	0.256 ***	[0.212, 0.298]	0.244 ***	[0.199, 0.288]	0.220 ***	[0.175, 0.266]
Reward -> DASS	−0.151 ***	[−0.199, −0.103]	−0.097 ***	[−0.146, −0.048]	−0.105 ***	[−0.157, −0.055]
Complexity -> DASS	0.046 *	[0.003, 0.089]	0.070 **	[0.025, 0.115]	0.033	[−0.014, 0.080]
Fate -> DASS	0.126 ***	[0.082, 0.171]	0.108 ***	[0.066, 0.152]	0.116 ***	[0.070, 0.161]
Religiosity -> DASS	−0.073 **	[−0.115, −0.030]	−0.023	[−0.068, 0.022]	−0.022	[−0.065, 0.022]

* *p* < 0.05. ** *p* < 0.01. *** *p* < 0.001. β = standardized regression coefficient; bootCI = 95% percentile bootstrapping confidence intervals with 5000 resamples for the standardized regression coefficient; DASS = Depression Anxiety Stress Scale.

**Table 5 ijerph-20-06224-t005:** The results of multiple regression on negative outcomes.

Variables	Depression			Anxiety			Stress		
	*b*	*SE(b)*	*CI*	*b*	*SE(b)*	*CI*	*b*	*SE(b)*	*CI*
Cynicism	0.195 ***	0.020	[0.154, 0.236]	0.216 ***	0.022	[0.170, 0.260]	0.196 ***	0.022	[0.148, 0.242]
Reward	−0.063 **	0.021	[−0.114, −0.011]	−0.066 **	0.023	[−0.118, −0.015]	−0.073 **	0.023	[−0.126, −0.019]
Complexity	0.036	0.019	[−0.006, 0.075]	0.056 **	0.022	[0.014, 0.100]	0.027	0.022	[−0.019, 0.075]
Fate Control	0.056 **	0.019	[0.015, 0.096]	0.070 **	0.022	[0.027, 0.114]	0.077 ***	0.022	[0.031, 0.121]
Religiosity	−0.021	0.019	[−0.062, 0.021]	0.002	0.022	[−0.043, 0.049]	0.010	0.022	[−0.036, 0.056]
MLQ-S	0.099 ***	0.019	[0.059, 0.139]	0.124 ***	0.021	[0.080, 0.168]	0.102 ***	0.022	[0.056, 0.147]
MLQ-P	−0.419 ***	0.021	[−0.464, −0.374]	−0.202 ***	0.023	[−0.248, −0.157]	−0.206 ***	0.023	[−0.254, −0.158]
Cynicism × MLQ-P	−0.071 ***	0.019	[−0.109, −0.031]	−0.032	0.021	[−0.071, 0.009]	−0.063 **	0.021	[−0.105, −0.019]
Reward × MLQ-P	0.008	0.018	[−0.038, 0.054]	0.006	0.020	[−0.035, 0.048]	0.008	0.020	[−0.039, 0.053]
Complex × MLQ-P	0.028	0.019	[−0.014, 0.070]	0.006	0.021	[−0.040, 0.050]	0.030	0.021	[−0.018, 0.075]
Fate × MLQ-P	−0.022	0.018	[−0.062, 0.018]	−0.014	0.020	[−0.054, 0.028]	0.004	0.020	[−0.038, 0.046]
Religiosity × MLQ-P	−0.011	0.018	[−0.053, 0.031]	−0.022	0.020	[−0.066, 0.021]	−0.044 *	0.020	[−0.085, −0.002]
Cynicism × MLQ-S	−0.008	0.018	[−0.051, 0.030]	−0.019	0.020	[−0.058, 0.019]	0.005	0.020	[−0.042, 0.048]
Reward × MLQ-S	0.027	0.017	[−0.020, 0.072]	0.039 ^a^	0.020	[−0.010, 0.084]	0.050 *	0.020	[−0.003, 0.103]
Complex × MLQ-S	0.007	0.017	[−0.034, 0.046]	−0.011	0.019	[−0.052, 0.032]	0.016	0.019	[−0.030, 0.062]
Fate × MLQ-S	−0.001	0.018	[−0.040, 0.038]	0.028	0.020	[−0.011, 0.067]	0.021	0.020	[−0.020, 0.063]
Religiosity × MLQ-S	−0.011	0.017	[−0.051, 0.029]	−0.012	0.019	[−0.056, 0.028]	−0.011	0.019	[−0.052, 0.032]

* *p* < 0.05. ** *p* < 0.01. *** *p* < 0.001. *b* = standardized regression coefficient; *SE(b)* = standard error of standardized regression coefficient; *CI* = 95% nonparametric bootstrapping confidence intervals. ^a^
*p* = 0.0505.

## Data Availability

The data used for this study will be made available by the corresponding authors upon request.

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
