# Peer review of "Meaning in Life, Social Axioms, and Emotional Outcomes during the First Outbreak of COVID-19 in Hong Kong"

_ijerph, 2023, doi:10.3390/ijerph20136224_

Round 1

Reviewer 1 Report

Some items are 7 liker scale and some are 5 liker scale, how could you justify they high variance and mean between the two ? The one deals with 7 likert scale will have probability of having high mean and variance asn compared to the other one. How will you justify it ?

Furthermore, negative emotions are using 4 point likert scale ?

You are discussing religious belief and meaning of life. Normally Chinese people think life is just for fun and there is no life here after. However, Muslims and Christians and Jews, beliefs on life here afer. This hope give them a meaning of life. I found some major lacking here, such as:

You didnt consider religious belief in your demographic variables.

You have used income as demographic variable, and through your discussiona and writting context shows that meaning of life is to earn money then you will be happy.

How do you justfiy the positive correlation between fate and depression, anxiety and stress.

Also now there is very popular term which is known as Eustress, or some people called Challenge stress, How do you see this stress in the context of your studies? Talk something about this stress and give your comments in the context of your studies ?

There is more than 25% plagiarism in your studies. Please bring it down to 20% or less.

Author Response

Some items are 7 liker scale and some are 5 liker scale, how could you justify they high variance and mean between the two ? The one deals with 7 likert scale will have probability of having high mean and variance asn compared to the other one. How will you justify it ? Furthermore, negative emotions are using 4 point likert scale ?

Response: Thank you for your comment. We used different Likert scales in our study because we used well-established questionnaires that had been adopted by psychometric studies with minimal modifications. For example, the DASS (Depression, Anxiety, Stress Scale) was validated by Lovibond and Lovibond (1995), and the original version used a scale of 0 to 3. The same applies to the other questionnaires we used. We adopted the Likert scales that had been used in the original questionnaires and established psychometrically before. This allowed us to compare our results with those of other studies.

You are discussing religious belief and meaning of life. Normally Chinese people think life is just for fun and there is no life here after. However, Muslims and Christians and Jews, beliefs on life here afer. This hope give them a meaning of life. I found some major lacking here, such as: You didnt consider religious belief in your demographic variables. You have used income as demographic variable, and through your discussiona and writting context shows that meaning of life is to earn money then you will be happy.

Response: Thank you for your comment. First of all, about religious belief, we talked about religiosity and the sample items are ‘Belief in a religion helps one understand the meaning of life’ and ‘Religion helps people make good choices for their lives’. We did not mention ‘religious belief”. Religiosity is not same as religious belief, it could be considered as people’s belief on religious belief. Second, even we used income as demographic variable, these are not our main variables and we did not suggest anything about income related to meaning in life in our paper. We just describe our sample. Third, to better describe our sample, we revised the manuscript and added one sentence about the description of religious belief of the sample: “Most participants identified as Atheist (70.01%)” and add detailed information in the Table 1 about participants’ religious beliefs. We hope that these revisions will help to clarify any misunderstandings and provide a more comprehensive description of our study.

How do you justfiy the positive correlation between fate and depression, anxiety and stress. Also now there is very popular term which is known as Eustress, or some people called Challenge stress, How do you see this stress in the context of your studies? Talk something about this stress and give your comments in the context of your studies ?

Response: Thank you for your comment. We explained this in Section 4.3: “Contrary to some previous studies such as Bachem et al. [3] and supported by other previous studies such as Kuo et al. [33], this study showed that fate control was associated with higher stress.” Later, we discussed our studies: “Under some circumstances, higher fate control might imply helplessness and desperation that one cannot change anything. Especially considering the data in this study were collected during the double-hit in Hong Kong, the negative correlation between fate control and presence might be a sign of the conflict between social beliefs (fate control) and existing personal beliefs (presence)…” In addition, stress in DASS defines stress as a negative emotional state. This measure did not measure eustress, though eustress could be an interesting topic. We emphasized negative emotional outcomes here. The example items for stress in DASS are “I found it hard to wind down,” “I tended to over-react to situations,” and “I felt that I was using a lot of nervous energy.”

There is more than 25% plagiarism in your studies. Please bring it down to 20% or less.

Response: Thank you for your comment. We have removed the sample items from the Methods section and paraphrased the definitions in the Literature Review section instead of using direct quotations. We believed the similarity rate should be lower than 20%.

Reviewer 2 Report

I like this paper. The topic is socially significant. I miss more "qualitative" approach and go beyond statistics. The article is highly quantitative and based on my experiences I believe that "the meaning of life" is a concept hard to present only with numbers. I think that the Authors (and readers) would benefit much more from the mixed methods approach and less "mechanical" research design.

Author Response

I like this paper. The topic is socially significant. I miss more "qualitative" approach and go beyond statistics. The article is highly quantitative and based on my experiences I believe that "the meaning of life" is a concept hard to present only with numbers. I think that the Authors (and readers) would benefit much more from the mixed methods approach and less "mechanical" research design.

Response: Thank you for your comment. We added one sentence in 4.4 Limitations and Future Direction: “Researchers could also consider to adopt mixed research design, which incorporates both quantitative and qualitative methods into the research to gain a deeper understanding of meaning in life.”

Reviewer 3 Report

A relevant topic is presented from a little-studied perspective, making the article original.

The structure of the introduction is appropriate, although I suggest expanding the explanation of the 5 dimensions of social axioms by providing an explanation of why the 3 social axioms (social complexity, reward for application, and fate control) have been little studied.

It would be ideal to include a figure that helps to understand the structure of the hypotheses.

The Cronbach's alphas resulting from the subscales of SAS-II for this sample are not high, a detailed explanation should be given as to why this scale was used.

The presentation of results and discussion is clear and well-founded.

Be careful with statements like “This study offered the first evidence of the mediating and moderating roles of the presence of and search for meaning in life on the relationships between social axioms and negative emotions, in a challenging time under the double-hit of COVID-19 pandemic following social unrest.” It is not good to assert so strongly because we cannot be sure that something has been published on the matter that has not been reviewed.

Overall, I consider this to be a good article, but specific issues need to be improved.

Author Response

A relevant topic is presented from a little-studied perspective, making the article original. The structure of the introduction is appropriate, although I suggest expanding the explanation of the 5 dimensions of social axioms by providing an explanation of why the 3 social axioms (social complexity, reward for application, and fate control) have been little studied.

Response: Thank you for your positive feedback. We added one sentence to explain the lack of studies regarding the three social axioms, stating that this is “due to the small effect size and conflicting results varying from one paper to another.” We also emphasized the importance of exploring these three social axioms, stating that “It is important to note that while the existing literature provides some insights about these three social axioms, there still remain many unanswered questions that warrant further research.”

It would be ideal to include a figure that helps to understand the structure of the hypotheses.

Response: Thank you for your comment. We added Figure 1 to illustrate our hypotheses.

The Cronbach's alphas resulting from the subscales of SAS-II for this sample are not high, a detailed explanation should be given as to why this scale was used.

Response: Thank you for your comment. We added two sentences to describe SAS-II in 2.2 Measure as following: “Previous studies have showed that this questionnaire has acceptable reliabilities ranging from .64 to .79, which are quite closed to this study [16]. This questionnaire was adopted mainly because its items were derived from Hong Kong participants and media sources [16].”

Be careful with statements like “This study offered the first evidence of the mediating and moderating roles of the presence of and search for meaning in life on the relationships between social axioms and negative emotions, in a challenging time under the double-hit of COVID-19 pandemic following social unrest.” It is not good to assert so strongly because we cannot be sure that something has been published on the matter that has not been reviewed.

Response: Thank you for your comment. In the abstract, we added “As far as we know, this study reported the first evidence …” and in the conclusion, we deleted the word “first”.